# Telecardiology in Rural Practice: Global Trends

**DOI:** 10.3390/ijerph19074335

**Published:** 2022-04-04

**Authors:** Takashi Yamano, Kazuhiko Kotani, Naomi Kitano, Junko Morimoto, Hiroki Emori, Masahiro Takahata, Suwako Fujita, Teruaki Wada, Shingo Ota, Keisuke Satogami, Manabu Kashiwagi, Yasutsugu Shiono, Akio Kuroi, Takashi Tanimoto, Atsushi Tanaka

**Affiliations:** 1Department of Cardiovascular Medicine, Wakayama Medical University, Wakayama 641-0012, Japan; morijun_men_kote@yahoo.co.jp (J.M.); hiroki_emori_wakayama@yahoo.co.jp (H.E.); masahiro.t72@gmail.com (M.T.); swkswk.414.414@gmail.com (S.F.); w_teruaki1026@yahoo.co.jp (T.W.); wakayama_hirosaki@yahoo.co.jp (S.O.); itsmeks14@gmail.com (K.S.); mkashi@wakayama-med.ac.jp (M.K.); yshiono@wakayama-med.ac.jp (Y.S.); akkuroi@gmail.com (A.K.); tmkktanimoto@gmail.com (T.T.); a-tanaka@wakayama-med.ac.jp (A.T.); 2Division of Community and Family Medicine, Jichi Medical University, Shimotsuke 329-0498, Japan; kazukotani@jichi.ac.jp; 3Health Administration Center, Wakayama Medical University, Wakayama 641-0012, Japan; naomiuk@wakayama-med.ac.jp

**Keywords:** rural and remote medicine, information and communication technology, telecardiology, tele-consultation, telemedical system, monitoring system, prehospital triage, tele-training

## Abstract

The management of cardiovascular diseases in rural areas is plagued by the limited access of rural residents to medical facilities and specialists. The development of telecardiology using information and communication technology may overcome such limitation. To shed light on the global trend of telecardiology, we summarized the available literature on rural telecardiology. Using PubMed databases, we conducted a literature review of articles published from January 2010 to December 2020. The contents and focus of each paper were then classified. Our search yielded nineteen original papers from various countries: nine in Asia, seven in Europe, two in North America, and one in Africa. The papers were divided into classified fields as follows: seven in tele-consultation, four in the telemedical system, four in the monitoring system, two in prehospital triage, and two in tele-training. Six of the seven tele-consultation papers reported the consultation from rural doctors to urban specialists. More reports of tele-consultations might be a characteristic of telecardiology specific to rural practice. Further work is necessary to clarify the improvement of cardiovascular outcomes for rural residents.

## 1. Introduction

Cardiovascular disease (CVD) is a leading cause of death worldwide [1]. Its prevention is therefore important. If a CVD event does occur, appropriate management should be implemented. For instance, acute coronary syndrome (ACS) is diagnosed by the electrocardiogram (ECG); meanwhile, the ability of doctors to analyze ECG is recognized to be variable according to their specialty [2]. Further, rural areas may lack cardiologists who are skillful at interpreting ECG findings. Patients with CVD living in rural areas have the problem of limited accessibility to not only cardiologists but also medical facilities. Indeed, patients with ACS in rural areas are reported to have difficulty in receiving reperfusion therapy [3,4,5].

Meanwhile, information and communication technology (ICT) has recently been applied to medical practice worldwide, and the applications (e.g., textual, auditory, and visual tools) show a wide range of utility [6]. Vollenbroek-Hutten et al. [6] reported that ICT offers a variety of opportunities for the treatment and prevention of frailty and functional decline in the aging society. However, he concluded that the actual use of ICT among the professionals is disappointingly low in contrast to the high use among patients. The development of telecardiology using ICT, which leads to travel and time reduction, may thus overcome the limitations of CVD management in rural practice [7,8]. Prehospital management of CVD [9] for rural residents would also be changed by telecardiology.

The global trends in telecardiology and merit investigation primarily to forecast the rural practice in CVD. However, no review has studied this topic. Thus, we aimed to summarize the global trends in telecardiology, which reduces rural disparities, enabling emergency management and follow up care, in rural practice.

## 2. Materials and Methods

### 2.1. Study Sites and Participants

For this review, experienced reviewers carried out the search to identify the titles and abstracts of potentially relevant studies [10]. Each abstract was assessed independently by two reviewers (T. Y. and K. K.). We used PubMed databases for the search of studies published from January 2010 to December 2020 in the English language using the following keywords: “rural telecardiology”, “rural tele cardiology”, “rural tele-cardiology”, “rural tele electrocardiogram”, and “rural tele ECG” (Figure 1). We selected total 21 original articles excluding reviews or meta-analyses and duplicate papers (rural telecardiology, 15 articles; rural tele cardiology, 11; rural tele-cardiology 3; rural tele electrocardiogram, 11; rural tele ECG, 12). If at least one of the reviewers considered a reference to be eligible, the article was obtained in its entirety. Both reviewers then independently analyzed the articles to select the ones to be included in the review. In the case of a disagreement, the decision was made by the authors’ consensus. Our study followed an interpretive analysis of the updated Preferred Reporting Items for Systematic reviews and Meta-Analyses (PRISMA) statement to promote evidence-based medicine [11]. We also performed a manual tracking of citations of the selected articles. We excluded two papers: one paper lacked information, and the other one paper reported on strokes with a clinical application perspective. Consequently, we identified 19 original papers eligible for the review.

### 2.2. Classification of Telecardiology

We classified the focus of the papers into tele-consultation, telemedical system telemedicine, monitoring system, prehospital triage, and tele-training [12,13,14,15], referring to the earlier literature on ICT. Tele-consultation [12] involves ICT-based communication to patients or caregivers (doctor to patient (D to P); D to P with nurse (D to P with N)) and informational exchange from doctors (i.e., general practitioners) to doctors (i.e., specialists; doctor to doctor (D to D)). A telemedical system [13] is a cloud system that provides remote transmission, interpretation, and storage of clinical parameters and useful diagnostic images to evaluate, diagnose, and treat patients at a distance using ICT. A monitoring system indicates the presence of a central monitoring center to which some physiological variables or images are sent [13]. Prehospital triage is defined as an emergent triage in the prehospital setting for the case of suspected ACS or arrhythmia [14]. Tele-training indicates education and teaching, such as the use of videoconference equipment, for medical staff delivered as remote lectures [15].

## 3. Results

Table 1 presents the 19 papers identified in our review, divided by geographical region (Asia, 9 papers; Europe, 7; North America, 2; Africa, 1). Seven papers focused on “tele-consultation” (numbers 2, 8, 12, 13, 16, 18, and 19 in Table 1), four papers focused on “telemedical system” (numbers 1, 5, 9, and 14), four papers focused on “monitoring system” (numbers 4, 7, 11, and 17), two papers focused on “prehospital triage” (numbers 3 and 6), and two papers focused on “tele-training” (numbers 10 and 15).

We analyzed the consultation type reported in the seven tele-consultation papers. In six papers, the consultation was from a rural doctor to an urban specialist (D to D); in one paper, (number 19), it was from a nurse in a nursing home to a general practitioner (D to P with N). We could not find any papers that reported D to P as a commonly used type of “telemedicine” or “telecare”.

Of the four papers on telemedical systems, three papers featured the creation of an overall structure using the cloud system. Only one paper (number 5) dissected the usage of tele-cardiology devices (via mobile phones). Of the four papers on monitoring systems, one paper (number 7) exhibited the technical system of transmission for the compressive detection of abnormal ECGs. Other studies discussed the advantage of portable ECGs for the detection of arrhythmia or heart failure. Meanwhile, the two studies on prehospital triage described the clinical benefits of ECG transfer in patients with ACS. Regarding tele-training, one study showed cardiovascular magnetic resonance (CMR) tele-training by fellowship, and one study presented tele-training on hearing heart murmurs in children.

## 4. Discussion

### 4.1. A New Summary

This review is the first to summarize the papers on telecardiology in rural areas. For rural residents with limited access to cardiologists and medical facilities, the development of telecardiology is expected to be helpful for better health care services [35]. The classification focus in this review included the fields of tele-consultation, telemedical system, monitoring system, prehospital triage, and tele-training. The classification and classified points might provide insights to the improvement of rural practice in CVD [36].

### 4.2. Tele-Consultation

Most of the papers in this review focused on tele-consultation. In rural areas with a limited accessibility to cardiologists and their medical facilities, ECG, echocardiography, or the other cardiac modality data need to be transferred immediately from the regional hospital to the city hospital. Thus, tele-consultation could be a characteristic of telecardiology specific to rural practice. Lazarus G. et al. [37] already reported that prehospital tele-ECG appeared to be an effective and worthwhile approach in the management of rural ACS patients on a systematic review because of decreased door-to-balloon time on catheter intervention. Moreover, this paper concluded that it may reduce the cardiovascular in-hospital and long-term mortality.

Otto et al. [17] reported on telehealth technology at the level of a rural community hospital emergency department in the U.S. They concluded that real-time tele-ultrasound consultation can serve as an important diagnostic resource in remote environments. Another study in India [33] discussed tele-emergency services at an altitude of over 3000 m. In the 35 months after tele-emergency training, 753 tele-consults were conducted between regional hospitals and community health centers, and several cases were transferred to larger centers or required helicopter transfer.

Ohligs et al. [34] reported on a holistic tele-consultation system for nursing home residents in a rural area in the first year of the COVID-19 pandemic. Although nurses are often inclined to call the rescue service for patients with acute symptoms, this tele-consultation system frequently applied video telephony, electrocardiography, and assessment of vitals for general practitioners. They concluded that the tele-consultation system, including integrated medical devices, was successfully developed. They also highlighted the utility of D to P with N teleconsultation and telemedicine in the context of a pandemic. In the future, D to P telemedicine may increase to reduce risky contact between patients and doctors.

### 4.3. Telemedical System

Three of the four papers focusing on telemedical systems reported that a cloud system could provide all patient information for health care staff as well as enable transparent checking of any data. They emphasize the importance of putting forward security solutions for each telemedical system and of centralizing all information on the cloud. The other paper in this classification [20] investigated the utility and wireless connection of mobile devices for a telemedical system developed by local young engineer. The authors suggested the usage of telehealth as a strategy to overcome the current health workforce shortage in South Africa, which will be ill-prepared to cope with the increasing demand for CVD care.

### 4.4. Monitoring System

One paper [22] presented the technical system of transmission of medical information. This paper proposed a two-tier framework enabling reliable resource-constrained telecardiology. The other three papers in this classification presented clinical cases of remote monitoring using mobile devices. Pineda-Lopz et al. [32] reported on a long-term Holter monitoring and 12-lead ECG recording system using mobile phones in Spain. They concluded that a 32-bit mobile microcontroller ECG system can be especially useful in rural areas of developing countries, where the lack of specialized medical personnel justifies the introduction of telecardiology services. A single-center study [26] in the U.S. reported on remote monitoring for heart-failure patients in underserved and rural communities, including Native American reservations. In this project, heart-failure patients with remote monitoring mobile devices showed substantial and statistically significant reductions in health care utilization compared with those who declined to participate. These findings indicated that a remote monitoring program using mobile devices can be successfully implemented in rural, underserved areas.

### 4.5. Prehospital Triage

Two studies focused on prehospital triage in patients with ACS. Brunetti et al. [18] reported a registry of tele-medicine services for patients with ST-segment elevation myocardial infarction in rural areas in Italy. Time to treatment in patients who received pre-hospital triage with tele-cardiology ECG was significantly shorter, and the rates of timely treatment were high. They concluded that pre-hospital triage with tele-cardiology ECG would be associated with shorter time to balloon and higher rates of timely treated patients even in rural areas. Büscher et al. [21] reported on a telemedical rescue assistance system in the German emergency medical services. This system can transmit audio and video data as well as vital signs and 12-lead ECG from the emergency site to a teleconsultation center. This article presented the impact of such a system on the use of telemedicine in rural areas. Given the importance of transferring simple data to the consultation center immediately in the emergent situation, Büscher et al. [21] focused on ECGs.

### 4.6. Tele-Training

Two papers focused on tele-training of medical staff. One study [25] reported a tele-training trial for CMR in Germany. In this study, network teaching for CMR reduced off-site training to only five weeks. CMR networks provide an efficient teaching platform with minimum off-site time for trainees in rurally located institutions. Tele-training involving screens, such as ECG and cardiac imaging, does not have to be in a face-to-face format. Although on-site training is important for acquiring clinical skills, part of it can be replaced by tele-training. For young doctors in rural areas, tele-training provides the opportunity to have clinical education without traveling to the city.

### 4.7. Additional Views

Experts have noted the number of initial and maintenance costs for the modalities to use telecardiology in the community [38]. Although all papers presented useful solutions for rural areas in their respective classified field (e.g., tele-consultation), we noted the need for a unified field (including tele-consultation, telemedicine, monitoring system, prehospital triage, and tele-training) based on comprehensive system, which would be more useful for the improvement of CVD treatment in rural areas. The government, rather than the community, should develop a comprehensive health care system that could be used anywhere, such as a cloud system. Such a system would not only reduce costs but also overcome geographical accessibility and save more lives.

### 4.8. Study Limitations

The present study had several limitations. First, this study included a relatively small number of reports because it was based on a literature review. Second, almost all studies used a cross-sectional design. The study periods of the studies with follow-up assessment were also short; thus, the effectiveness of telecardiology on outcomes (i.e., CVD-related mortality) for rural residents remains undetermined. Third, few studies reported the cost and infrastructure of telecardiology, which represents an important issue in rural practice. Lastly, while newer technologies, such as artificial intelligence (AI) that is implied to improve medical practice [39], may be already introduced to rural practice, we could not find such a paper on the telecardiology combined with AI in rural practice. This is a limitation in the review based on the previously published literature.

## 5. Conclusions

In this review, the trend suggests the global use of telecardiology in rural practice. We found that tele-consultation was the focus of interest in more papers compared with other fields, which might be a characteristic of telecardiology specific to rural practice. Further work is necessary to clarify whether the use of telecardiology improves the diagnosis and treatment of CVD for rural residents.

## Figures and Tables

**Figure 1 ijerph-19-04335-f001:**
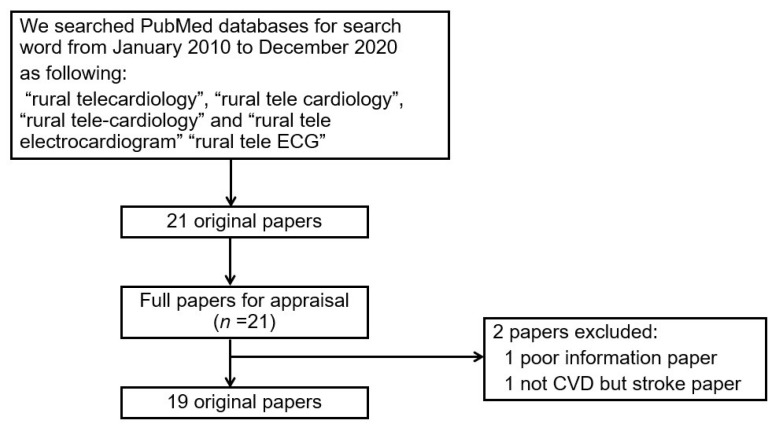
The flowchart of the search for papers. ECG, electrocardiogram; CVD, cardiovascular disease.

**Table 1 ijerph-19-04335-t001:** Papers on rural telecardiology included in the review.

No.	Author (Reference)	Year	Country (Continent)	Equipment	Main Findings	Classified Field	Usage Method and Used Population	Clinical Outcome
1	Hsieh et al. [16]	2012	Taiwan (Asia)	A new cloud and pervasive computing-based 12-lead ECG telemedical service	The inclusion of new cloud service in ECG telemedical service, an ECG tele-diagnosis via cellphone, and cloud-based 12-lead ECG e-learning; upgrades previously developed applications, facilitates the collaboration of hospitals, and enhances the efficacy of telemedical system.	Telemedical system	Network among ambulance, clinic, and hospital by cloud system	Diagnosis on ECG
2	Otto et al. [17]	2012	USA (North America)	Telehealth consultation at the level of a rural community hospital	Real-time tele-ultrasound consultation can serve as an important diagnostic resource in remote environments.	Tele-consultation	Rural doctor to urban specialist by tele-ultrasound consultation	Diagnosis on echocardiology
3	Brunetti et al. [18]	2014	Italy (Europe)	A registry of prehospital triage of patients with ST-segment elevation myocardial infarction	Time to treatment in patients who received prehospital triage with the tele-cardiology ECG is significantly shorter, and rates of patients timely treated are higher.	Prehospital triage	Ambulance to catheterization laboratory for primary PCI by tele-medicine service	Treatment of ACS
4	Singh et al. [19]	2014	India (Asia)	Clinical validation of handheld (portable) tele-ECG as a screening tool	The tele-ECG method shows similar results with conventional ECG, with 99% correlation. Tele-ECG is a portable, cost-effective, and convenient tool for monitoring heart diseases.	Monitoring system	Patient to remote centers by portable tele-ECG	Diagnosis on ECG
5	Noubiap et al. [20]	2014	South Africa (Africa)	Telemedical strategies for using a Cardionet system in developed locally	A tele-cardiology device provides several advantages in terms of cost, ease of use, autonomy, and reduced technology requirements. Test results are transferred wirelessly via mobile phone connection to specialist physicians who can interpret them and provide assistance with case management.	Telemedical system	Network among cardiologist, remote hospital, and remote center by Cardionet system (Cloudnet system)	Diagnosis on ECG
6	Büscher et al. [21]	2014	Germany (Europe)	Telemedical rescue assistance system in German emergency medical services	Telemedical rescue assistance system can transmit audio and video data as well as vital signs and 12-lead ECG from the emergency site to a teleconsultation center for patients with ACS.	Prehospital triage	Ambulance to teleconsultation center by telemedical rescue assistance system	Treatment of ACS
7	Chandra et al. [22]	2015	India (Asia)	Two-tier framework enabling reliable resource-constrained telecardiology with limited power and bandwidth in rural communities	High reliability is maintained even at substantial power and bandwidth savings in a monitoring system in India.	Monitoring system	ECG signal transmission from user end to diagnostic center (mainly technical system, population not specified)	Improvement of technical system
8	Lapão et al. [23]	2015	Portugal (Europe)	International tele-consultation services in supporting the evacuation procedures from Africa to Europe	This study provides evidence (e.g., case study, interview, cost reduction) of the importance of telemedicine for coping with both geographical constraints and shortage of physicians.	Tele-consultation	Rural doctor to urban specialist by international telemedicine service	Diagnosis of heart disease
9	De la Torre-Diez et al. [24]	2015	Spain (Europe)	Cloud system called eHealth Services in Spanish rural health centers	This telemedical system provides all information on the cloud in patients at a local area for health care staff and they could transparently check any data.	Telemedical system	Network among rural health centers by electronic health records (cloud)	Diagnosis of heart disease
10	Muehlberg et al. [25]	2015	Germany (Europe)	Tele-training trial for cardiovascular magnetic resonance	Network teaching reduced off-site training to only five weeks and provided an efficient teaching platform with a minimum of off-site time for trainees in rurally located institutions.	Tele-training	Fellows teach trainees by module-based network	Diagnosis on magnetic resonance
11	Riley et al. [26]	2015	USA (North America)	Remote monitoring for heart failure patients in a single center	Heart-failure patients with remote monitoring mobile devices showed substantial and statistically significant reductions in health care utilization compared with those who declined to participate.	Monitoring system	Monitor for heart-failure patients by broadband-enabled remote monitoring devices	Diagnosis of Heart failure
12	Nagayoshi et al. [27]	2016	Japan (Asia)	Benefit and utility of a Digital Imaging and Communications in Medicine tele-consultation network at a rural hospital	Ten cases (20.8%) of 48 tele-consultations that had been conducted were transferred to a high-volume center. The tele-consultation network enabled open communication between distant hospitals.	Tele-consultation	Rural low-volume hospitals to high-volume centers by broadband network	Network on DICOM
13	Shetty et al. [28]	2017	India (Asia)	Feasibility of tele-consultation to link rural clinics to a teaching hospital	ECGs were transmitted to the hospital with 99.7% success on first attempt. The staff at the hospital were able to provide timely interpretation of ECGs and advice to patients.	Tele-consultation	Rural clinics to city hospital by tele-ECG system	Diagnosis on ECG
14	De la Torre-Diez et al. [29]	2017	Spain (Europe)	Cloud-based telemedical system solution in a hospital, a health center in a city, and health centers in a rural area	ICT software simulated different scenarios to provide an adapted solution in the form of a telemedical service.	Telemedical system	Rural hospital to city health center by mobile-ECG telemedicine application	Diagnosis of heart disease
15	Pyles et al. [30]	2017	China (Asia)	A system to allow rural physicians to obtain assistance in the diagnosis and management of children with heart disease	The project tested the hypothesis that acceptable screening of heart murmurs could be accomplished using a digital stethoscope and internet cloud transmittal to deliver phonocardiograms to an experienced observer. The overall test accuracy was 91% with 78.5% sensitivity and 92.6% specificity.	Tele-training	From remote clinic to regional cardiology center by HeartLink tele-auscultation system (tablet and computer)	Diagnosis of heart disease in children
16	Cauhan et al. [31]	2018	India (Asia)	Whether to reduce the time taken for diagnosis of acute coronary syndrome using 24 h tele-ECG-consultation support	The hospital-to-aspirin time of the tele-consultations group was significantly reduced compared with the control group. This is an effective, low-cost, and replicable strategy.	Tele-consultation	Primary care physicians to specialist physician by tele-ECG system	Treatment of ACS
17	Pineda-Lopz et al. [32]	2018	Spain (Europe)	12-lead ECG recording and monitoring system using mobile phones	The 32-bit mobile microcontroller ECG system can be especially useful in rural areas in developing countries.	Monitoring system	Monitor for patient’s 12-lead/Holter data by cellphone and webserver	Diagnosis of heart disease
18	Ganapathy et al. [33]	2019	India (Asia)	24 h tele-emergency consultation services at an altitude over 3000 m	753 teleconsults were given in the first 35 months, and several cases were transferred to larger centers or required helicopter transfer.	Tele-consultation	Regional hospital to community health center by tele-emergency services	Diagnosis of heart disease
19	Ohligs et al. [34]	2020	Germany (Europe)	A holistic tele-consultation system for nursing home residents	This telemedical system applied video telephony, electrocardiography, and assessment of vitals for general practitioners	Tele-consultation	Nurses in nursing homes to general practitioner by exclusive telemedical system	Diagnosis of heart disease

ECG, electrocardiogram; PCI, percutaneous coronary intervention; ACS, acute coronary syndrome; DICOM, Digital Imaging and Communications in Medicine.

## Data Availability

The data presented in this study are available upon request from the corresponding author.

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
