# Peer review of "Telecardiology in Rural Practice: Global Trends"

_ijerph, 2022, doi:10.3390/ijerph19074335_

Round 1

Reviewer 1 Report

In Table 1.... Please write First author et al

In References  Please check

Standardise pages of periodic ...compare ref 2 and 4

The name of periodic in ref 10 and volume in ref 17 

and in ref 36 Nov 1 and fascicle are dispensable

In INTRODUCTION, the authors could emphasize more deeply the advantages and disadvantages provided by the use of information and communication technology (ICT), that are of potential interest to face the challenges of our aging society (ref 6. Vollenbroek-Hutten, M.; Jansen-Kosterink, S.; Tabak, M.; Feletti, L.C.; Zia, G.; N'dja, A.; Hermens, H.; SPRINTT Consortium. Possibilities of ICT-supported services in the clinical management of older adults. Aging Clin Exp Res. 2017, 29 :49-57. doi: 230 10.1007/s40520-016-0711-6).

In CONCLUSIONS, authors could emphasize some comments on Artificial intelligence (AI). AI is making computer systems capable of executing human brain tasks in many fields in all aspects of daily life. The enhancement in information and communications technology (ICT) has indisputably improved the quality of people's lives around the globe. For example, cardiovascular imaging has now accurate imaging combined with big data from the eHealth record and pathology to better characterize the disease and personalized therapy (Yan Cheng Yang et al. Influential Usage of Big Data and Artificial Intelligence in Healthcare. Comput Math Methods Med. 2021 ; 2021:5812499. doi: 10.1155/2021/5812499. eCollection 2021).

Author Response

Dear Reviewer 1:

We thank the Reviewer for his time and input.

We found the mistake for the institution, so corrected. (line 12)

Reviewer’s comment 1

In Table 1.... Please write First author et al

Author’s response 1

According to your advice, we add First author et al. (table1)

Reviewer’s comment 2

Standardise pages of periodic ...compare ref 2 and 4

The name of periodic in ref 10 and volume in ref 17

and in ref 36 Nov 1 and fascicle are dispensable

Author’s response 2

We corrected Standardise pages in all references, the name of in ref 10 and the volume in ref 17 and delete Nov 1 in ref 36.

Reviewer’s comment 3

In INTRODUCTION, the authors could emphasize more deeply the advantages and disadvantages provided by the use of information and communication technology (ICT), that are of potential interest to face the challenges of our aging society (ref 6. Vollenbroek-Hutten, M.; Jansen-Kosterink, S.; Tabak, M.; Feletti, L.C.; Zia, G.; N'dja, A.; Hermens, H.; SPRINTT Consortium. Possibilities of ICT-supported services in the clinical management of older adults. Aging Clin Exp Res. 2017, 29 :49-57. doi: 230 10.1007/s40520-016-0711-6).

Author’s response 3

Thanks for your meaningful advice. We stated in the Introduction section as follows: “Vollenbroek-Hutten, M et al. [6] has reported that ICT offers a variety of opportunities for the treatment and prevention of frailty and functional decline in the aging society. However, he problematized that the actual use of ICT among the professionals is disappointingly low in contrast to the high use among patients.” (Line 45-49)

Reviewer’s comment 4

In CONCLUSIONS, authors could emphasize some comments on Artificial intelligence (AI). AI is making computer systems capable of executing human brain tasks in many fields in all aspects of daily life. The enhancement in information and communications technology (ICT) has indisputably improved the quality of people's lives around the globe. For example, cardiovascular imaging has now accurate imaging combined with big data from the eHealth record and pathology to better characterize the disease and personalized therapy (Yan Cheng Yang et al. Influential Usage of Big Data and Artificial Intelligence in Healthcare. Comput Math Methods Med. 2021 ; 2021:5812499. doi: 10.1155/2021/5812499. eCollection 2021).

Author’s response 4

I agree with you about the importance of AI for improving the quality of people's lives in the near future, but we can detect no paper that suggested mainly the relationship with the telecardiology at rural and AI in this review because 19 original papers which were identified for this review have been reported before 2021.

We added the limitation in the discussion as follows; “Lastly, while newer technologies, such as artificial intelligence (AI) that is implied to improve medical practice [40], may be already introduced to rural practice, we could not find such a paper on the telecardiology combined with AI in rural practice. This is a limitation in the review based on the previously published literature.” (Line 216-219)

And we added the reference:

  1. Yang, Y.; Islam, S.U.; Noor, A.; Khan, S.; Afsar, W.; Nazir, S. Influential Usage of Big Data and Artificial Intelligence in Healthcare. Comput Math Methods Med. 2021, 5812499. doi: 10.1155/2021/5812499. eCollection 2021

Reviewer 2 Report

Rural clinicians must act quickly to achieve optimal patient outcomes. New technologies (tele ECG, tele Echo, mobiles, prehospital triage) enable better access to rapid diagnosis, as well as specialist input and care of cardiovascular disease, including acute coronary syndromes and life-frightening arrhythmias.

The presented paper is a systematized review of 19 cross-sectional studies on the rural tele cardiology. Both introduction and methods are complete and exhaustive; very clear and on point.

Results could be more focused on the cardiovascular outcomes. I think it would be easier to read Table 1 if it was somehow systematized, e.g. with headings: acute coronary syndromes, arrhythmias, ultrasonography, patients follow-up, etc.

Also, I think that the discussion could address some more data and needs enriching references. Many Authors found that tele-ECG may reduce door-to-balloon (DTB) time as well as the cardiovascular in-hospital and long-term mortality. This issues are well described in the review paper by Lazarus G et al. Prehospital telecardiology improves managements of acute coronary syndrome in rural populations: a systematic review and meta-analysis. J Telemed Telecare. 2020 Sep 30;1357633X20960627. doi: 10.1177/1357633X20960627. Online ahead of print.

The Authors might also emphasize that the main objective of these actions is to reduce rural disparities, enabling emergency management, and follow up care in rural contexts.

Author Response

Dear Reviewer 2:

We thank the Reviewer for his time and input.

We found the mistake for the institution, so corrected. (line 12)

Reviewer’s comment 1

Results could be more focused on the cardiovascular outcomes. I think it would be easier to read Table 1 if it was somehow systematized, e.g. with headings: acute coronary syndromes, arrhythmias, ultrasonography, patients follow-up, etc.

Author’s response 1

According to your advice, we add the project of “clinical outcome”. (table1)

Reviewer’s comment 2

Also, I think that the discussion could address some more data and needs enriching references. Many Authors found that tele-ECG may reduce door-to-balloon (DTB) time as well as the cardiovascular in-hospital and long-term mortality. This issues are well described in the review paper by Lazarus G et al. Prehospital telecardiology improves managements of acute coronary syndrome in rural populations: a systematic review and meta-analysis. J Telemed Telecare. 2020 Sep 30;1357633X20960627. doi: 10.1177/1357633X20960627. Online ahead of print.

Author’s response 2

Thanks for your meaningful advice. We stated in the discussion section as follows and reference section:

“Lazarus G et al. [38] has already reported that prehospital tele-ECG appeared to be an effective and worthwhile approach in the management of rural ACS patients on a sys-tematic review because of decreased door-to-balloon time on catheter intervention. Moreover, this paper concluded that it may reduce the cardiovascular in-hospital and long-term mortality. ” (line131-135)

Reference:

  1. Lazarus, G.; Kirchner, H.L.; Siswanto, B.B. Prehospital tele-electrocardiographic triage improves the management of acute coronary syndrome in rural populations: A systematic review and meta-analysis. J Telemed Telecare. 2020, 1357633X20960627. doi: 10.1177/1357633X20960627. Online ahead of print.

Reviewer’s comment 3

The Authors might also emphasize that the main objective of these actions is to reduce rural disparities, enabling emergency management, and follow up care in rural contexts.

Author’s response 3

We changed the last sentence in the Introduction section as follows: “Thus, we aimed to summarize the global trends in telecardiology, which reduces rural disparities, enabling emergency management, and follow up care, in rural practice. ” (line 54-57)